# Influence of Genotype on High Glucosinolate Synthesis Lines of *Brassica rapa*

**DOI:** 10.3390/ijms22147301

**Published:** 2021-07-07

**Authors:** Prabhakaran Soundararajan, Sin-Gi Park, So Youn Won, Mi-Sun Moon, Hyun Woo Park, Kang-Mo Ku, Jung Sun Kim

**Affiliations:** 1Genomics Division, Department of Agricultural Bio-Resources, National Institute of Agricultural Sciences, Rural Development Administration, Wansan-gu, Jeonju 54874, Korea; prabhu89@korea.kr (P.S.); soyounwon@korea.kr (S.Y.W.); msoutlaw@korea.kr (M.-S.M.); hwpark0803@korea.kr (H.W.P.); 2Bioinformatics Team of Theragen Etex Institute, Suwon 16229, Korea; singi.park@theragenbio.com; 3BK21 Interdisciplinary Program in IT-Bio Convergence System, Chonnam National University, Gwangju 61186, Korea; ku9@chonnam.ac.kr; 4Department of Horticulture, Chonnam National University, Gwangju 61186, Korea

**Keywords:** biologically active compounds, glucosinolate synthesis, recombinant blocks, doubled haploid lines, isothiocyanates, cancer prevention

## Abstract

This study was conducted to investigate doubled haploid (DH) lines produced between high GSL (HGSL) *B**rassica rapa* ssp. *trilocularis* (yellow sarson) and low GSL (LGSL) *B. rapa* ssp. *chinensis* (pak choi) parents. In total, 161 DH lines were generated. GSL content of HGSL DH lines ranged from 44.12 to 57.04 μmol·g^−1^·dry weight (dw), which is within the level of high GSL *B. rapa* ssp. *trilocularis* (47.46 to 59.56 μmol g^−1^ dw). We resequenced five of the HGSL DH lines and three of the LGSL DH lines. Recombination blocks were formed between the parental and DH lines with 108,328 single-nucleotide polymorphisms in all chromosomes. In the measured GSL, gluconapin occurred as the major substrate in HGSL DH lines. Among the HGSL DH lines, BrYSP_DH005 had glucoraphanin levels approximately 12-fold higher than those of the HGSL mother plant. The hydrolysis capacity of GSL was analyzed in HGSL DH lines with a Korean pak choi cultivar as a control. Bioactive compounds, such as 3-butenyl isothiocyanate, 4-pentenyl isothiocyanate, 2-phenethyl isothiocyanate, and sulforaphane, were present in the HGSL DH lines at 3-fold to 6.3-fold higher levels compared to the commercial cultivar. The selected HGSL DH lines, resequencing data, and SNP identification were utilized for genome-assisted selection to develop elite GSL-enriched cultivars and the industrial production of potential anti-cancerous metabolites such as gluconapin and glucoraphanin.

## 1. Introduction

There is experimental evidence supporting the idea that compounds in Cruciferous plants are effective against cancer and heart ailments [1,2,3]. Strong anti-carcinogenic effects of members of the family Brassicaceae are attributed to their glucosinolate (GSL) content [4,5]. For consumption, some GSLs such as glucoraphanin (GRA), glucoalyssin (GAL), gluconapin (GNA), neoglucobrassicin (NGBS), and gluconasturtiin (GNT) are considerably beneficial whereas hydrolysis products from progoitrin (PRO), epiprogoitrin (epiPRO), and gluconapoleiferin (GNL) can cause goiter in animals [6]. GSLs are not actual bioactive agents, but rather their hydrolysis products such as isothiocyanates (ITCs), nitriles, epithionitriles, thiocyanates, and indoles [7]. Some of the important bioactive compounds are butenyl sulforaphane (SFN), isothiocyanate (BITC) and pentenyl isothiocyanate (PEITC) (Figure 1) [6,7].

The biosynthesis of GSL is one of the more complex processes and more than 130 GSLs have been identified so far [8]. The existence of the different GSL structures is controlled by variability in genes on individual genotypes, especially at loci involved in initial elongation and side-chain modification reactions [9]. In addition, the molecular function of a gene can be altered depending on the plant species, allelic condition, and polymorphic state of the regulatory network controlling it [10]. According to previous works, four major loci, *GS-ELONG*, *GS-OX*, *GS-AOP* (*GS-ALK* and *GS-OHP*), and *GS-OH,* control the difference in the accumulation of aliphatic GSLs [6,11,12]. *GS-ELONG* consists of genes methylthioalkylmalate synthase 1 (*MAM1)*, *MAM2*, and *MAM3* to regulate the chain length of GSLs [13,14]. The *GS-OX* loci contain *flavin monooxygenase* (*FMO_GS-ox_*) and *GS-AOP* two tightly linked loci, *GS-ALK* for *2-oxoglutarate-dependent dioxygenases* (*AOP2*) and *GS-OHP* for *AOP3* involved in the side-chain modification reaction determines the type of GSL products [15,16,17]. *AOP2* genes are responsible for the conversion of GRA to GNA. In most *B rapa*, GNA makes up the major proportion of total GSL. However, the presence of stop codons in *B**. oleracea* leads *BoAOP2.2* and *BoAOP2.3* into non-functional genes. This results in the major GSL content in *B. oleracea* being GRA instead of GNA [18,19,20]. *GS-OH* locus controls oxidation of 3-butenyl GSL to alkenyl GSL [6,21]. Most importantly, the accumulation and content of GSL are strongly influenced by R2R3-myeloblastosis (MYB) transcription factors (TFs) [22]. *MYB28*, *MYB**29*, and *MYB**76* regulate the aliphatic GSL genes while *MYB34*, *MYB**51*, and *MYB**122* regulate the indolic GSL genes [23]. For fine mapping and selection of individual genotypes, it is necessary to identify allelic discrepancies on key loci and for different TFs.

Resequencing provides the opportunity to develop a vast amount of novel markers. Identification of genes related to important agronomic traits, genetic diversity analysis, characterize and environmental factors influences are mandatory in genome-assisted breeding for crop improvement [24]. Combining quantitative trait loci (QTL) mapping with whole-genome sequencing helps in the precise detection of functional loci for traits of interest and their candidate genes. Major advantages of resequencing are functional allele mining and single-nucleotide polymorphism (SNP) discovery between large populations. The identification of trait loci in biparental cross populations by high-resolution linkage map and functional allele mining utilizing SNPs and Insertions–Deletions (InDels) as markers provides a powerful complementary strategy to genome-wide association studies (GWAS) [25]. Recently, Chen et al. (2016) resequenced 199 and 119 accessions representing 12 and nine morphotypes of *B. rapa* and *B. oleracea*, respectively. This resequencing data aided in the identification of leaf-heading and tuberous morphotypes associated with sub-genome parallel selection during diversification and domestication of *B. rapa* and *B. oleracea* [26]. Meanwhile, the resequencing of 588 *B. napus* accessions led to the identification of A and C sub-genome origins [27]. Similarly, genome mapping helped to characterize genes responsible for club root resistance [28], blackleg resistance [29], black rot resistance [30], and flowering time [31], etc.

*B. rapa* is one of the model plants for GSL metabolism and polyploid species with phenotypically diverse cultivated subspecies. In our previous study, we determined that eight subspecies of *B. rapa* have different GSL levels ranging from 4.42 μmol g^−1^ dry weight (dw) in *B. rapa* ssp. *narinosa* to 53.51 μmol·g^−1^·dw in *B. rapa* ssp. *trilocularis* [32]. The current study was conducted to identity SNPs based on resequencing between high GSL (HGSL) and low GSL (LGSL) content doubled haploid (DH) lines generated from two different parents, *B. rapa* ssp. *trilocularis* (yellow sarson) and *B. rapa* ssp. *chinensis* (pak choi). Yellow sarson is an oil plant with relatively high amounts of several beneficial GSLs, whereas pak choi is one of the major leafy vegetables in Korean and Chinese diets but contains lesser amounts of GSL. The aim of this work is to produce edible DH lines with high beneficial GSLs and low toxic substances like PRO. Detailed analysis of recombinant blocks between the HGSL and LGSL lines was carried out based on GSL biosynthetic pathways. The results from this study could be useful for precise genome-assisted selection in the development of elite *B. rapa* cultivars with enriched GSL content for commercial purposes.

## 2. Results

### 2.1. Generation of BrYSP DH Lines and GSL Content Profiles

Accessions YS-033 (CGN06835) and PC-099 (CGN132924) [33] were maintained in our greenhouse. As neither of these accessions were perfect inbred genotypes, we performed selfing for four generations (S_4_) of YS-033 and named LP08. DH plant of PC-099 was named LP21. LP08 and LP21 were used as the set of homozygous parents. The F_1_ plant was developed using a female of LP08 and male of LP21 (Figure 2). Microspore cultures were followed according to our previous method [34,35]. The leaf edge shapes were segregated in female and male parent’s phenotype boundary (Figure 2, middle). Phenotype classification of the leaf edge shape consisted of (a) entire, (b) slightly serrated, (c) intermediately serrated, and (d) very serrated. Our population was named “BrYSP_DH000” in accordance with *B. rapa* (Br), yellow sarson + pak choi (YSP), doubled haploid (DH), and a unique three-digit number for each line. The regulation of GSL biosynthesis was then examined in Chiffu, LP08, LP21, F_1_, and 161 BrYSP_DH plants.

We selected five HGSL lines and three LGSL lines among the 161 BrYSP_DH plants for further studies, including resequencing and GSL analysis (Table 1, Figure 2). Total GSL amounts were detected from 44.12 ± 2.86 μmol·g^−1^·dw (BrYSP_DH005) to 57.04 ± 1.54 μmol·g^−1^·dw (BrYSP_DH026). It was consistent with the range of total GSL for *B. rapa* ssp. *trilocularis*, a higher GSL content subspecies in our previous study (47.46 to 59.56 μmol g^−1^ dw) [32]. The most abundant GSL in the HGSL lines was GNA, which was 80 to 91% of the total GSL. We selected the HGSL lines based on lesser PRO content (<0.6 μmol g^−1^ dw) as PRO is known to cause harmful effects on human health. As shown in Table 1, the BrYSP_DH005 line exhibited a highly unique profile with enriched aliphatic GSLs GRA (1.23 ± 0.18 μmol g^−1^ dw) and GAL (3.47 ± 0.18 μmol g^−1^ dw), indolic GSL NGBS (2.03 ± 0.20 μmol g^−1^ dw), and aromatic GSL GNT (0.68 ± 0.34 μmol g^−1^ dw). Significance difference between the DH lines was observed in all GSLs (Table 1).

### 2.2. Resequencing of Parents, HGSL Lines, and LGSL Lines

Sequencing reads of 150 bp were generated using 10 paired-end sequencing libraries of 350 bp insert size. Resequencing of LP08 and LP21 produced around 775 and 704 million raw reads, respectively. The total bases of sequence obtained were 78.2 Gb for LP08 and 71.1 Gb for LP21. The sequences of the DH lines ranged from around 125.5 to 192.4 million clean reads. These reads sequenced were mapped to the *B. rapa* v3.0 reference genome [36]. Mapping reads (%) between parents differed with 90.65% (LP08) and 93.19% (LP21). The DH lines had slightly higher and more similar mapping read percentages, ranging from 94.8% to 95.98%. The genome coverages for the two parent sequences had a 157.05× depth for LP08 and 156.07× depth for LP21. Nine DH line were mapped from 46.89× (BrYSP_DH059) to 70.80× (BrYSP_DH026) (Table 2).

### 2.3. SNP Genotyping and InDels

Approximately ~3.5 million and ~2.5 million SNPs were predicted in LP08 and LP21 lines in reference to *B. rapa* v3.0 genome, respectively [36]. SNP density was calculated as 27.5 per 1-kb in LP08 and 22.2 per 1-kb in LP21. The maximum number of InDels were identified in LP08, i.e., 724,760. Comparatively, a lesser number of InDels were detected in LP21, i.e., 552,220. On average, 2,707,400 high-quality SNPs and 617,408 InDels were identified in the DH lines. Details regarding the SNPs and InDels of the parents and DH lines are provided in Appendix A.

### 2.4. Identification of GSL Biosynthesis-Specific Recombinant Blocks

Overall, 108,328 variants were extracted as allelic differences between LP08 and LP21 in the recombinant block search of 342 recombinant blocks among the genotypes (Appendix A, Appendix A). Out of 110 GSL biosynthetic genes, 75 were identified in recombinant blocks and mapped to their respective chromosomes (Figure 3). Uniformly, the recombinant blocks between the region of 12,929,867–23,248,122 bp of A03 with 3716 SNPs were discovered and identified to be present in only the HGSL lines (Table 3). The ten GSL synthesis genes positioned between 12.9 Mb and 23.2 Mb of A03 commonly differed between all resequenced HGSL and LGSL lines. In detail, TFs *MYB28.1* of aliphatic GSLs and *MYB34.2* of indolic GSLs of the HGSL lines were LP08 types. Similarly, for the chain elongation step, *branched-chain amino acid aminotransferase 4* (*BCAT4*; *Bra001761*), *MAM1* (*Bra013007*), *MAM3* (*Bra013009* and *Bra013011*), and *bile acid transporter 5* (*BAT5; Bra000760*) of the HGSL lines belong to LP08. *AOP1* (*Bra000847*) and *AOP2* (*Bra000848*) were the two keys genes derived from LP08 in high GSL lines for the side-chain modification process. One of the genes involved in the sulfur donation from chloroplast to sulfotransferase for production of desulfo GSL, *APS*
*kinase* (*APK*)*1* (*Bra013120*) is the LP08 type in all high GSL lines (Table 3).

Though all *MAM* genes are LP08 type in BrYSP_DH005, only three *MAM* genes are LP21 types in other high DH lines. In a similar way, all *MAM* genes were LP21 in BrYSP_DH059 and BrYSP_DH061 but three genes were LP08 type in the BrYSP_DH009 line. Even though in the core structure synthesis phase of BrYSP_DH005 most of the genes were identified as LP21 type, other HGSL lines showed about 50% of genes were LP08 type. It is noteworthy that BrYSP_DH014, BrYSP_DH026 and BrYSP_DH016, BrYSP_DH017 shared similar parental type recombinant blocks of GSL biosynthetic genes.
ijms-22-07301-t002_Table 2Table 2Summary of re-sequencing and alignment result for DH lines of *Brassica rapa*.Sample IDNo. of ReadsNo. of BasesNo. of Clean ReadsClean Reads (%)No. of Clean BasesClean Bases (%)De-Duplicated ReadsDe-Duplicated Reads (%)MappedReadsMapped Reads (%)Ave. Coverage (x)LP08(♀)775,036,33278,278,669,532748,357,76696.5675,085,735,554 95.92613,028,55481.92555,700,09490.65157.05LP21(♂)704,684,56271,173,140,762676,201,11895.9667,699,611,150 95.12592,219,74387.58551,893,41293.19156.07BrYSP_DH005135,586,70820,473,592,908132,434,32097.6819,674,335,422 96.10121,145,70491.48116,215,95995.9348.48BrYSP_DH009130,566,11019,715,482,610128,070,23298.0919,054,916,880 96.65120,327,98593.95115,494,19495.9848.06BrYSP_DH014143,278,51221,635,055,312141,074,21698.4621,128,122,142 97.66133,925,20094.93127,330,97795.0853.5BrYSP_DH016134,052,43420,241,917,534132,201,87098.6219,784,969,829 97.74123,574,66693.47117,994,49895.4849.38BrYSP_DH017145,561,74821,979,823,948144,029,95298.9521,590,666,966 98.23133,547,33192.72126,848,29994.9853.23BrYSP_DH026194,633,90429,389,719,504192,401,31698.8528,832,473,186 98.1176,347,87391.66168,647,64495.6370.80BrYSP_DH059126,815,04219,149,071,342125,484,03298.9518,829,119,983 98.33117,116,77493.33111,549,89395.2546.89BrYSP_DH061138,437,79420,904,106,894137,010,40698.9720,534,954,754 98.23127,865,42493.33122,305,76795.6551.45
ijms-22-07301-t003_Table 3Table 3Gene blocks identified as high (LP08) and low (LP21) GSL parents with the DH lines.






♀HGSL LineLGSL Lines♂ChromStartStopNameGene ID V1.5Gene ID V3.0StagesLP08DH005DH014DH016DH017DH026DH009DH059DH061LP21**A03****21326869****21328218*****MYB28.1******Bra012961******BraA03g044440.3C***Transcription factors—AliphaticLP08LP08LP08LP08LP08LP08LP21LP21LP21LP21A0934691133470483*MYB28.2**Bra035929**BraA09g007000.3C*LP08LP21LP08LP08LP08LP08LP21LP21LP08LP21A022540387525405492*MYB28.3**Bra029311**BraA02g043310.3C*LP08LP08LP21LP21LP21LP21LP08LP21LP21LP21A0313643091365719*MYB29.1**Bra005949**BraA03g003070.3C*LP08LP08LP08LP21LP21LP08LP21LP21LP08LP21A0933564333357581*MYB34.1**Bra035954**BraA09g006760.3C*Transcription factors—IndolicLP08LP21LP08LP08LP08LP08LP21LP21LP08LP21**A03****21129279****21130669*****MYB34.2******Bra013000******BraA03g043850.3C***LP08LP08LP08LP08LP08LP08LP21LP21LP21LP21A022518382625184998*MYB34.3**Bra029349**BraA02g042890.3C*LP08LP08LP21LP21LP21LP21LP08LP21LP21LP21A022517290625179858*MYB34.4**Bra029350**BraA02g042880.3C*LP08LP08LP21LP21LP21LP21LP08LP21LP21LP21A081824835218249890*MYB51.2**Bra016553**BraA08g028300.3C*LP08LP21LP08LP08LP08LP08LP08LP21LP21LP21A0668416886842966*MYB51.3**Bra025666**BraA06g013940.3C*LP08LP21LP21LP08LP08LP21LP08LP08LP21LP21A072341131323412768*MYB122.1**Bra015939**BraA07g037950.3C*LP08LP08LP08LP08LP08LP08LP08LP08LP08LP21A021232734912329278*MYB122.2**Bra008131**BraA02g022140.3C*LP08LP08LP21LP21LP21LP21LP08LP21LP08LP21**A03****18302919****18305368*****BCAT-4****Bra001761**BraA03g039030.3C*Side chain elongationLP08LP08LP08LP08LP08LP08LP21LP21LP21LP21A051830515218307632*BCAT-4**Bra022448**BraA05g027600.3C*LP08LP21LP08LP08LP08LP08LP08LP08LP08LP21A0688492378851230*BCAT-3**Bra017964**BraA06g017190.3C*LP08LP21LP21LP08LP08LP21LP08LP08LP21LP21A011481488914817183*BCAT-3**Bra029966**BraA01g023550.3C*LP08LP21LP21LP21LP21LP21LP21LP21LP21LP21A0223916912393547*IPMDH1**Bra023450**BraA02g005400.3C*LP08LP21LP21LP21LP21LP21LP08LP21LP08LP21**A03****21085392****21093219*****MAM1****Bra013007**BraA03g043770.3C*LP08LP08LP08LP08LP08LP08LP21LP21LP21LP21**A03****21061112****21065885*****MAM3****Bra013009**BraA03g043760.3C*LP08LP08LP08LP08LP08LP08LP21LP21LP21LP21**A03****21054537****21056963*****MAM3****Bra013011**BraA03g043750.3C*LP08LP08LP08LP08LP08LP08LP21LP21LP21LP21A021593474415944413*MAM1**Bra018524*
LP08LP08LP21LP21LP21LP21LP08LP21LP21LP21A022511550725119702*MAM1**Bra029355**BraA02g042820.3C*LP08LP08LP21LP21LP21LP21LP08LP21LP21LP21A022510344925106649*MAM3**Bra029356**BraA02g042810.3C*LP08LP08LP21LP21LP21LP21LP08LP21LP21LP21A0456538745657118*IPMI LSU1**Bra032708**BraA04g008730.3C*LP08LP21LP21LP21LP21LP21LP21LP21LP21LP21A0859640645967370*IPMI LSU1**Bra040341**BraA08g006340.3C*LP08LP21LP21LP08LP08LP21LP08LP21LP21LP21A0516083811609157*IPMI SSU2**Bra004744**BraA05g003360.3C*LP08LP21LP21LP08LP08LP21LP21LP08LP21LP21**A03****12943140****12945015*****BAT5****Bra000760**BraA03g027810.3C*LP08LP08LP08LP08LP08LP08LP21LP21LP21LP21A091720446817206516*BAT5**Bra029434**BraA09g026220.3C*LP08LP08LP08LP08LP08LP08LP08LP08LP21LP21A0660079266010112*CYP79F1**Bra026058**BraA06g012170.3C*Core structure synthesis—AliphaticLP08LP21LP21LP08LP08LP21LP08LP08LP21LP21A0454603935462018*CYP83A1**Bra032734**BraA04g008410.3C*LP08LP21LP21LP21LP21LP21LP21LP21LP21LP21A02768271770517*CYP79A2–Aromatic**Bra028764**BraA02g001710.3C*LP08LP21LP21LP21LP21LP21LP08LP21LP08LP21A052481169824812553*GSTF11**Bra032010**BraA05g041750.3C*LP08LP21LP08LP08LP08LP08LP08LP21LP08LP21A071749841817499341*GSTU20**Bra003645**BraA07g026570.3C*LP08LP08LP08LP08LP08LP08LP08LP08LP08LP21A0133596203360837*GGP1**Bra011201**BraA01g007200.3C*LP08LP21LP21LP21LP21LP21LP21LP21LP08LP21A032790199827903448*GGP1**Bra024068**BraA03g029390.3C*LP08LP08LP08LP21LP21LP08LP21LP21LP21LP21A081283345112834884*GGP1**Bra010282*
LP08LP21LP21LP08LP08LP21LP08LP21LP21LP21A081283583012837549*GGP1**Bra010283**BraA08g017720.3C*LP08LP21LP21LP08LP08LP21LP08LP21LP21LP21A0959496185952361*SUR1**Bra036703**BraA09g011980.3C*LP08LP21LP08LP08LP08LP08LP08LP21LP08LP21A092512365825125223*UGT74B1**Bra024634**BraA09g038870.3C*LP08LP08LP08LP08LP08LP08LP08LP08LP21LP21A011696487816965828ST5b*Bra031476**BraA01g028650.3C*LP08LP21LP21LP21LP21LP21LP21LP21LP21LP21A071847954618480592ST5b*Bra003817*
LP08LP08LP08LP08LP08LP08LP08LP08LP08LP21A071848174818482782ST5b*Bra003818**BraA07g028360.3C*LP08LP08LP08LP08LP08LP08LP08LP08LP08LP21A071800272918003723ST5b*Bra003726**BraA07g027400.3C*LP08LP08LP08LP08LP08LP08LP08LP08LP08LP21A072341904823420082ST5b*Bra015938**BraA07g037960.3C*LP08LP08LP08LP08LP08LP08LP08LP08LP08LP21A072342659523427635ST5b*Bra015936**BraA07g037980.3C*LP08LP08LP08LP08LP08LP08LP08LP08LP08LP21A0977267677727825ST5b*Bra027623**BraA09g012830.3C*LP08LP08LP08LP08LP08LP08LP08LP08LP08LP21A0990193119020375ST5b*Bra027117**BraA09g016490.3C*LP08LP08LP08LP08LP08LP08LP08LP08LP08LP21A0990216419022747ST5b*Bra027118**BraA09g016500.3C*LP08LP08LP08LP08LP08LP08LP08LP08LP08LP21A0999778929978911ST5b*Bra027880**BraA09g017830.3C*LP08LP08LP08LP08LP08LP08LP08LP08LP21LP21A0668500536851066ST5c*Bra025668**BraA06g013960.3C*LP08LP21LP21LP08LP08LP21LP08LP08LP21LP21A01374137375874*CYP79B2**Bra011821**BraA01g000840.3C*Core structure synthesis—IndolicLP08LP21LP08LP08LP08LP08LP21LP21LP08LP21A033120316931205210*CYP79B2**Bra017871**BraA03g061480.3C*LP08LP08LP08LP21LP21LP08LP21LP21LP21LP21A081497344414975600*CYP79B2**Bra010644**BraA08g021670.3C*LP08LP21LP08LP08LP08LP08LP08LP21LP21LP21A0850580975059748*CYP83B1**Bra034941**BraA08g007380.3C*
LP08LP21LP21LP08LP08LP21LP08LP21LP21LP21A0372304007231268*GSTF9**Bra022815**BraA03g016240.3C*
LP08LP08LP08LP08LP08LP08LP21LP21LP08LP21A041390144213902299*GSTF9**Bra021673**BraA04g022220.3C*
LP08LP21LP08LP21LP21LP08LP21LP21LP21LP21A0372328447233982*GSTF10**Bra022816**BraA03g016250.3C*
LP08LP08LP08LP08LP08LP08LP21LP21LP08LP21A021233718012338199*ST5a**Bra008132**BraA02g022150.3C*
LP08LP08LP21LP21LP21LP21LP08LP21LP08LP21A072342790923428469*ST5a**Bra015935*

LP08LP08LP08LP08LP08LP08LP08LP08LP08LP21A081920381019205763*FMOGS-OX5**Bra016787**BraA08g030720.3C*Side chain modification—AliphaticLP08LP21LP08LP08LP08LP08LP08LP21LP21LP21A0982178248219479*FMOGS-OX2**Bra027035**BraA09g015600.3C*LP08LP08LP08LP08LP08LP08LP08LP08LP08LP21A021597374115976278*AOP2**Bra018521**BraA02g027430.3C*LP08LP08LP21LP21LP21LP21LP08LP21LP21LP21**A03****13492936****13494533*****AOP1****Bra000847**BraA03g028760.3C*LP08LP08LP08LP08LP08LP08LP21LP21LP21LP21**A03****13498815****13503183*****AOP2****Bra000848**BraA03g028770.3C*LP08LP08LP08LP08LP08LP08LP21LP21LP21LP21A0911706261172022*AOP1**Bra034182**BraA09g001250.3C*LP08LP21LP08LP08LP08LP08LP21LP21LP08LP21A0911685031169786*AOP1**Bra034181**BraA09g001260.3C*LP08LP21LP08LP08LP08LP08LP21LP21LP08LP21A0911650281166807*AOP2**Bra034180*
LP08LP21LP08LP08LP08LP08LP21LP21LP08LP21A0377682937769612*GSL-OH**Bra022920**BraA03g017350.3C*LP08LP08LP08LP08LP08LP08LP21LP21LP08LP21A041385255113853809*GSL-OH**Bra021670**BraA04g022180.3C*LP08LP21LP08LP21LP21LP08LP21LP21LP21LP21A041387739713878646*GSL-OH**Bra021671**BraA04g022190.3C*LP08LP21LP08LP21LP21LP08LP21LP21LP21LP21A0258463925848174*CYP81F2**Bra020459**BraA02g012540.3C*Side chain modification—IndolicLP08LP08LP21LP21LP21LP21LP08LP21LP08LP21A0352339245236233*CYP81F2**Bra006830**BraA03g012390.3C*LP08LP08LP08LP08LP08LP08LP21LP21LP08LP21**A03****20376817****20378290*****APK1****Bra013120**BraA03g042630.3C*
LP08LP08LP08LP08LP08LP08LP21LP21LP21LP21A01361733363288*APK2**Bra011822**BraA01g000800.3C*Sulphur supplementationLP08LP21LP08LP08LP08LP08LP21LP21LP08LP21A033121896931220359*APK2**Bra017872**BraA03g061500.3C*LP08LP08LP08LP21LP21LP08LP21LP21LP21LP21A081498024814981406*APK2**Bra010645**BraA08g021680.3C*LP08LP21LP08LP08LP08LP08LP08LP21LP21LP21Bold letters are indicated when all HGSL lines had the LP08 genotypes whereas LGSL lines had the LP21 genotypes. Pink (LP08) and sky-blue (LP21).


### 2.5. Comparative Analysis of GSL Pathway between Individual Genotypes

Stepwise comparative analysis on the GSL biosynthetic pathway between the HGSL line BrYSP_DH005 and LGSL line BrYSP_DH059 was performed based on the recombinant blocks of parents LP08 and LP21. Key differences were observed between BrYSP_DH005 and BrYSP_DH059 in amino acid elongation and side-chain modification step TFs such as *MYB28* and *MYB29*. Entire *MAM1*, *AOP1*, *AOP2*, and *GSL-OH* genes of BrYSP_DH059 were present as LP21 recombinant blocks. In contrast, complete *MAM1*, one *AOP1* (*Bra000847*), two *AOP2* (*Bra018521* and *Bra000848*), and one *GSL-OH* (*Bra022920*) were LP08 blocks in BrYSP_DH005. Interestingly, all genes of indolic and aromatic pathways in BrYSP_DH059, except *ST5a* (*Bra024634*) and *UGT74B1* (*Bra024634*), were LP21 recombination blocks. Contrastingly, critical genes for the synthesis of glucobrassicin (GBA) in BrYSP_DH005, such as *ST5a* (*Bra008132* and *Bra015935*), and *CYP81F2* (*Bra020459* and *Bra006830*), were LP08 blocks. Except for *MYB28.2* (*Bra035929*), TFs involved in the biosynthesis of the aliphatic GSL were LP08 blocks in BrYSP_DH005, whereas, aliphatic TFs of BrYSP_DH059 were LP21 blocks. A similar trend was observed in *MYB34* TFs. Other than *MYB34.2* (*Bra013000*), all *MYB34* homologues in BrYSP_DH005 were LP08 types. *MYB34* TFs in BrYSP_DH059 remained as the LP21 genotype (Figure 4).

### 2.6. GSL Hydrolysis Products

Main hydrolysis products of GSL such as BITC, 4-PEITC, 2-PEITC, and SFN of high GSL lines were compared with the commercial pak choi cultivar used in South Korea. All five of the representative HGSL DH lines possessed increased amounts of hydrolysis products. Overall, BrYSP_DH014 had the highest level of hydrolysis products (870.29 µg∙g^−1^ dw). It is about 6.3-fold higher levels than that of the commercial cultivar. The HGSL line with the lowest level of hydrolysis products was the BrYSP_DH017 line which is about 417.5 µg∙g^−1^ dw. Still, it had a 3.0-fold high-level hydrolysis product than that of the control pak choi. Most importantly, the amount of anti-carcinogenic agent SFN in BrYSP_DH005 was 20.2 µg∙g^−1^ dw. This was 7–10-fold higher than that in the other HGSL DH lines and nearly 35-times more than that of the commercial cultivar (Table 4).

### 2.7. Nitrile Formation

Among the five HGSL DH lines tested, BrYSP_DH005 showed significantly lower nitrile formation compared to that of the other HGSL DH lines, from both SNG and GNT substrate-based assays (Figure 5). The lower SNG substrate-based nitrile formation may explain why BrYSP_DH005 generated higher concentrations of SFN compared to other HGSL DH lines (Table 4). Although a decrease in GNT-based nitrile formation was observed in BrYSP_DH005 compared with other HGSL DH lines, it was more increased than the commercial pak choi.

## 3. Discussion

In this study, the representative lines were selected from previously developed 161 DH lines with HGSL and LGSL content for metabolite, resequencing, SNP mapping, and GSL biosynthetic pathway analysis. Total GSL content ranged from 44 μmol·g^−1^·dw to 57 μmol·g^−1^·dw in HGSL DH lines (Table 1). Amounts of GSL were significantly higher than those in the inbred line *Brassicaraphanus* “BB1” [7] and broccoli (*B. oleracea* var. italica) [38]. Glucoraphasatin is the major GSL of *Raphanus sativus* [39] and *Brassicoraphanus* “BB1” [7], SNG in mustard [1] and horseradish [40] cultivars. Similar to our study, GNA was found to be the predominant GSL in *B. rapa* “Chinese cabbage”. However, the range is around 54% only [41], but in our study, GNA is 80% to 91% of total GSL. Though GNA is the major GSL present in HGSL DH lines, BrYSP_DH005 possessed a considerable amount of other important metabolites such as GRA, GAL, NGBS, and GNT (Table 1). Due to its higher content of several beneficial GSLs, BrYSP_DH005 was selected as a vital genotype and detailed analysis on the GSL pathway was carried out in comparison with the representative LGSL line BrYSP_DH059 (Table 1).

For any hybrid studies, SNPs and InDels are valuable for developing linkage maps of trait loci in the cross-population lines. High-density SNPs and InDels polymorphism markers could be a valuable resource for genetic-linkage studies and precise QTL mapping of desirable traits [24,42] in *Brassica* (Appendix A, Appendix A). The predicted recombinant block between 12.9–23.2 bp of A03 chromosome in the GSL-rich parent, all five HGSL lines, and three LGSL lines indicate that it is a key region that should be focused on GWAS (Table 3, Figure 3). GWAS analysis in combination with metabolite profiling has gained widespread acceptance to assess natural variations between populations [43].

Sønderby et al. (2010) noted that decreases in a few transcripts can have a major impact on the accumulation of GSL [22]. Epistatic effects of transcript expression levels are highly complex to mirrored with the GSL content. Resequencing followed by recombinant block analysis showed a clear picture of genes derived from HGSL and LGSL parents (Appendix A, Figure 3). The results of this study indicate that integration of critical recombinant blocks from parents LP08 and LP21 triggered to turn on the genes involved in biosynthesis, transportation and regulation in the GSL metabolic pathway (Figure 2, Figure 3 and Figure 4).

Transcriptional activation of aliphatic and indolic genes by *MYB* factors are as follows: *MYB28*, *MYB29* and *MYB76* activate *MAM*, *CYP79F1/F2*, *CYP83A1*, *GGP1*, *SUR1*, *UGT74B1/C1*, and *STb/c* of the aliphatic pathway and *MYB34*, *MYB51*, and *MYB 122* activate *CYP79B2/B3*, *CYP83B1*, *YGT74B1*, and *STa* of the indolic pathway [23]. MYB factors also modulate genes involved in the sulfur supplementation route such as *ATPS1*/*S3* and *APK1*/*2* [44]. For instance, *MYB28.2*, *MYB 34.1* and *MYB51.2* are LP08 derived in all HGSL DH lines except BrYSP_DH005. Contrastingly, *MYB28.3*, *MYB34.3*, *MYB34.4* and *MYB122.2* were LP08 types in BrYSP_DH005 whereas they are LP21-derived in other HGSL lines. This could be correlated to the excess amounts of GRA, GAL, NGBS, and GNT and also to the lesser content of GNA, GBN, and 4-MOGBS in BrYSP_DH005 than other HGSL lines. Similarly, the LP08 type of *MYB29.1* in BrYSP_DH005, BrYSP_DH014, and BrYSP_DH026 can be associated with an appreciable amount of GAL (Table 1 and Table 3). Induction of glucoraphasatin biosynthesis genes by MBY29 in radish (*R. sativus* L.) root was reported [39]. LP08-type recombinant block with *MYB29.1* could also be related to a higher level of GSL hydrolysis products (Table 3 and Table 4). Though MYBs involved in aliphatic and indolic pathways are close homologs to each other, they have distinct functional roles [45]. However, their functions overlap and they are able to substitute for each other in cases of mutants, decreased expression, and overexpression [22,45,46,47].

The first chain elongation process catalyzed by *BCAT4* deaminates Met and homoMet to corresponding 2-oxoacids. Two copies of *BAT5* serve as the importer of 2-oxo acids into plastid. Out of two copies of *BCAT4* and one copy (*Bra001761*) present in all HGSL lines belonged to the LP08 genotype. The other copy (*Bra022448*) was also LP08-type in all HGSL DH lines except BrYSP_DH005. Both copies of *BAT5* were LP08-type in BrYSP_DH005. Individual mutants of *bcat4* and *bat5* exhibit approximately a 50% reduction in the level of aliphatic GSLs [48]. As the *GS-ELONG* locus contains *MAM* genes, the number of side chains of GSL is decided by the elongation cycles it undergoes the initial steps [14]. Gene duplication, neo-functionalization, and polymorphism of *MAM1* lead to the diversification of GSL profiles [49]. Following isomerization by *isopropylmalate* (*IPM*) *isomerase* and oxidative decarboxylation by *IPM dehydrogenase* (*IPM-DH*), the 2-oxo acid yields homoMet and chain-elongated derivatives of Met to enter the core GSL structure pathway [6].

There are five genes present in the *GS-OX* locus of *A. thaliana* (*FMO_GS-ox1-5_*), but only one copy of *FMO_GS-ox2_* and two copies of *FMO_GS-ox5_* have been identified in *B. rapa*. This locus is responsible for the oxygenation of glucoibervirin, glucoerucin, and glucoberteroin based on the co-expression of TFs for aliphatic GSLs [15,16]. *AOP2* belongs to *GS-ALK* for conversion of S-oxygenated GSLs and *AOP3* belongs to *GS-OHP* for conversion of hydroxyl GSLs [6]. Our fine-mapping on the key locus regions such as *BCAT4*, *MAM1*, *BAT5*, *AOP2*, and *GS-OH* in recombinant inbred lines showed the high production of GNA with less conversion of PRO (Figure 4, Table 1, Table 3 and Appendix A).

*Cytochrome P450* (*CYP*) *CYP79F1* catalyzes the conversion of all chain-elongated Met derivatives. *CYP79F1* mainly for long-chained Met derivatives, but no copies of *CYP79F2* have been found in *B. rapa* [50]. *CYP79B2* and *CYP79B3* are identified for indolic whereas *CYP79A2* for aromatic derivatives [51]. The resulting aldoximes are then converted into nitrile oxides or *aci*-nitro compounds by *CYP83A1* for Met derivatives and *CYP83B1* for Trp and Phe derivatives [52]. Glutathione-S-transferase (GSTF) is involved in the catalysis of nitrile into S-alkyl-thiohydroximate. In this step, sulfur is supplied to alkyl-thiohydroximate as glutathione (GSH) [53]. GSH is a tripeptide with a gamma peptide linkage between glutamate and cysteine with glycine. S-alkyl thiohydroximates are converted into thiohydroximates by four copies of *Y**-glutamyl peptidase 1* (*GGP1*) and two copies of *superroot1* (*SUR1*) [54]. S-glucosylated is catalyzed by glucotransferase. *UDP-glucosyl transferase 74C1* (*UGT74C1*) and *UGT74B1* metabolize Met-derived and Phe-derived compounds of thiohydroximates into desulfo GSL [55]. In the final step of core structure biosynthesis, the sulfate donor 3′-phosphoadenosine-5′-phosphosulfate (PAPS) is produced by two steps. First, ATP sulfurylase (ATPS) produces the intermediate adenosine-5′-phosphosulfate (APS). APS kinase then catalyzes APS to Cys. Of four copies of *APK1*, *Bra013120* present in all the HGSL were LP08 type. The other three copies varied among the HGSL DH lines (Table 3). All *APK1* and *APK2* genes were LP21 type in BrYSP_DH059 (Figure 4b). Double mutants of *apk1* and *apk2* reduce total GSL in *A. thaliana* by almost 80%. In addition, the accumulation of desulfoGSL is noticeable in mutants, but present at undetectable levels in wild-type plants [56]. Core structures of GSLs depend on sulfur assimilation, especially sulfotransferase (ST) reaction [9,55]. Two copies of *ST5a* catalyze both Trp and Phe desulfoGSL. About ten copies of *ST5b* and one copy of *ST5c* are involved in the synthesis of aliphatic GSLs, such as glucoibervirin, glucoerucin, and glucoberteroin [57].

Hydrolysis of GSLs by endogenous myrosinase (β-D-thioglucosidase) produces active compounds, such as ITCs, nitriles, and indoles [3]. An important ITC, SFN is reported to be a natural inducer of phase II detoxification enzymes, including glutathione-S-transferase and quinone reductase (QR). SFN triggers cytostasis and apoptosis and also detoxifies xenobiotics [7]. BITC and PEITC have proved to induce apoptosis in cancer cell lines [8,9]. Nitriles have a weaker chemopreventive effect than ITCs [7]. Although broccoli is a well-known health-promoting vegetable due to high SFN concentrations, it has a wide range of ESP activity from 17.1% to 46% among 20 commercial broccoli cultivars [58]. In contrast, the HGSL DH lines of the current study exhibited much lower nitrile formation, and increasing GRA levels may have directly contributed to the induction of phase II detoxification enzymes (Figure 6). Nitrile formation using SNG as a substrate in the HGSL DH lines ranged from 0.44% to 2.66%, which was considerably lower than nitrile formation compared with 11 commercial mustard cultivars (7.4–62.4%) or USDA fancy horseradish (average, 7.1%) [1,40]. Although nitrile formation based on GNT of BrYSP_DH005 was significantly higher than commercial pak choi, it was still lower than the recent report on broccoli [38]. In this study, important ITCs including SFN, BITC, 4-PEITC, and 2-PEITC were the major hydrolysis compounds along with some nitrile hydrolysis products from GNA (1-cyano-3,4-epithiobutane) and GNT (benzenepropanenitrile) (Appendix A). Our current multilayered analysis of resequencing and the revelation of SNP-based recombinant block discovery results will be helpful for further fine QTL mapping.

This information will be beneficial to the production of elite GSL-enriched cultivars for commercialization of potential anti-cancerous metabolites, such as GRA, GLA, NGBS, and GNT, with higher SFN activity.

## 4. Materials and Methods

### 4.1. Plant Materials

Accessions YS-033 (CGN06835) and PC-099 (CGN132924) [33] were provided by professor Guusje Bonnema. We performed selfing for four generations (S_4_) of YS-033 and named LP08. DH plant of PC-099 was named LP21. LP08 and LP21 were used as the set of homozygous parents. The F_1_ plant was developed using an LP08 as female and LP21 as male parent (Figure 1). Microspores were collected and cultures were followed according to our previous method [34,35]. *B. rapa* ssp. *perkinensis* “Chiffu-401-42” was used as the reference.

### 4.2. HPLC Analysis for Identification of GSL Content

GSL content was estimated according to Seo et al. [32]. Fresh leaves of 6-weeks old plants were freeze-dried and 100 mg samples were used for protein extraction by boiling with 1.5 mL of 70% (*v*/*v*) methanol in a 10 mL test tube for 10 min at 95 °C. Extracts were loaded on Sephadex A25 columns and desulfation was conducted with aryl sulfatase (EC 3.1.6.1) before HPLC. Desulfated GSLs were quantified in Agilent 1200 Series HPLC System (Agilent Technologies, Santa Clara, CA, USA) equipped with an Inertsil ODS-3 column (150 × 3.0 mm inner diameter, particle size 3 µm; GL Science, Tokyo, Japan). Analysis was done using a flow rate of 0.4 mL·min^−1^ at a column over temperature of 35 °C and a wavelength of 227 nm. Sinigrin (SNG) was used as an external standard for quantification. Total and individual GSL content was calculated as means of three biological replicates.

### 4.3. Resequencing

Genomic DNA was extracted from fresh leaves as previously described [59]. In liquid nitrogen, 5 g of samples were finely ground and put in 50 mL falcon tube. It is mixed with the pre-warmed 15 mL of DNA extraction buffer (500 mM NaCl, 100 mM Tris-HCL, pH 8.0, 50 mM EDTA, pH 8.0, 1.25% SDS, and add 0.38% sodium bisulfite before use with adjust pH 8.0 with 0.2 N, NaOH). Incubate for 30 min, and invert gently every ten min at 65 °C. Add 5 mL of aquaphenol and rotate at 14 rpm for 10 min at room temperature (RT). Add to an equal volume of Chloroform: Isoamyalchole (24:1) and rotate at 20 rpm for 15 min at RT. Centrifuge for 15 min at 15 °C at 10,000 rpm. Supernatant was transferred to a new 50 mL falcon tube. Add 10 µL of RNaseA (20 mg/mL) and incubate at 37 °C in 10 min. An equal proportion of isopropanol was added and gently mix by inverting. Carefully pull out the DNA pellet with the closed sterile glass tube. DAN pellet was washed with 70% ethanol several times. Completely dry the pellet under vacuum pressure. Finally, dissolve the pellet in 0.1× TE buffer. Libraries with an average insert size of 350 bp were constructed using the genomic DNA TruSeq Nano DNA Sample Prep Kit according to the manufacturer’s protocol. Sequencing was performed with 150 bp paired-end sequencing using the NovaSeq6000 platform. Reads were converted from the binary base call (BCL) format file using bcl2fastq V2.20 software with parameter “–barcode-mismatches 0”.

### 4.4. Variant Calling

To remove low-quality bases, clean reads were trimmed using sickle v1.33 with a Phred quality score threshold of 20 (accuracy 99%) to derive high-quality reads. The mapped reads were aligned against the *B. rapa* reference genome sequence available in the Brassica database (BRAD V3.0, http://brassicadb.org/brad/ (accessed on 1 December 2020) using mem algorithm in Burrows–Wheeler Aligner (BWA) V0.7.17 software “-Am”, “-k 45”. After alignment, duplicates were marked using MarkDuplicates tools in GATK V4.0.2.1. Multi-sample variant calling of SNPs and InDels were performed using Haplotype Caller in GATK V4.0.2.1.

### 4.5. Identification of Recombinant Blocks

For recombinant block search, extraction of variants with allelic differences between LP08 and LP21 was performed. Recombinant blocks for each sample were defined as LP08, LP21, or hetero based on SNPs and InDels using in-house scripts. GSL biosynthetic genes were identified using BLASTp with E-value of 1 × 10^−5^ against the *B. rapa* reference genome [36]. BioSample accession number of DH lines deposited in NCBI are as follows: BrYSP_DH005, SAMN18818129; BrYSP_DH009, SAMN18818130; BrYSP_DH014, SAMN18818131; BrYSP_DH016, SAMN18818133; BrYSP_DH017, SAMN18818134; BrYSP_DH026, SAMN18818135; BrYSP_DH059, SAMN18818139; BrYSP_DH061, SAMN18818140.

### 4.6. Quantification of Glucosinolate Hydrolysis Products

Freeze-dried sample powder (50 mg) was suspended in 2 mL micro-centrifuge tube (Fisher Scientific, Waltham, MA, USA) with 1 mL distilled water. Under darkness for 24 h, hydrolysis products were generated naturally by endogenous myrosinase. Samples were added with 1 mL of dichloromethane and centrifuged at 12,000× *g* for 2 min. Lower organic layer was carefully collected. To quantify the GSL hydrolysis products, gas chromatograph (GC) (6890N, Agilent Technologies) coupled to an MS detector (5975B, Agilent Technologies) equipped with an auto sampler (7683B, Agilent Technologies) and a capillary column (30 m × 0.32 mm × 0.25 µm J&W HP-5, Agilent Technologies) was used. From extract, 1 µL was injected in GC-MS with split ratio of 1:1. Initial temperature was set to 40 °C for 2 min, then the oven temperature was increased to 260 °C at 10 °C/min and hold for 10 min. Temperature of injector and detector were set at 200 °C and 280 °C, respectively, with the flow rate in the helium carrier at 1.1 mL/min. Peaks were identified using the respective standards [40,41].

### 4.7. Measurement of Nitrile Formation

Nitrile formation (%) was measured to estimate the epithiospecifier protein (ESP) activity as ESP enhances the formation of nitriles over isothiocyanates. Nitrile formation in each sample was determined by incubating concentrated horseradish root extract with crude protein extract of the sample and analyzed using gas chromatography–mass spectrometry (GC-MS). Firstly, concentrated horseradish extract was prepared by powdered 10 g root samples were mixed in 100 mL of 70% methanol. After centrifuging at 4000× *g* for 5 min, supernatant was boiled in the beaker until all solvent was evaporated and reconstituted in 50 mL of deionized water. Freeze-dried powder (75 mg) of *B. rapa* DH leaves sample were mixed with 1.5 mL of concentrated “1091” horseradish root extract in 2 mL microcentrifuge. Centrifugation was carried out at 12,000× *g* for 2 min. Supernatant (0.6 mL) was transferred to 1.5 mL Teflon centrifuge tube (Savillex Corporation, Eden Prairie, MN, USA) and mixed with 0.6 mL of dichloromethane. Samples were incubated in RT for 1 h upside down to minimize volatile compounds loss. Vortexed tubes were centrifuged at 12,000× *g* for 2 min. Bottom organic layer was injected to GC-MS (Trace 1310 GC, Thermo Fisher Scientific, Waltham, MA, USA) coupled to a MS detector system (ISQ QD, Thermo Fisher Scientific, Waltham, MA, USA) and an auto sampler (Triplus RSH, Thermo Fisher ScientificA capillary column (DB-5MS, Agilent Technologies; 30 m × 0.25 mm × 0.25 µm capillary column). The sample was held at 40 °C for 2 min. Oven temperature was increased to 320 °C at 15 °C/min and held for 4 min. Injector and detector temperature were set at 270 °C and 275 °C, respectively. Flow rate of helium carrier gas was 1.2 mL/min. Standard curve was used to quantify hydrolysis rate of nitriles [40,41].

### 4.8. Statistical Analysis

Analysis of variance (ANOVA) was performed using SAS Enterprise Guide 7.1 (SAS Institute Inc., Carrey, NC, USA). Tukey’s honest significant difference (HSD) test was performed using Prism 5 software (GraphPad, San Diego, CA, USA).

## 5. Conclusions

The present study advances our knowledge regarding the inheritance of GSL biosynthetic genes in *B. rapa* for high GSL synthesis and increased ITCs with low concentrations of nitriles. The result of this work will be useful for genome-assisted precise breeding in *B. rapa*. Metabolite profiling in various DH lines and its recombinant block predicted in A03 (12.9 Mb-23.2 Mb) chromosome based on SNPs and InDels broadens our understanding of GSL biosynthesis and the key genes responsible for the production of beneficial GSL. The integrative genetic-linkage map brings detailed knowledge of variants in GSL biosynthetic genes in the A03 chromosome region. Further recombinant blocks responsible for GSL biosynthesis can be used for the selection and development of GSL-rich edible cultivars of *B. rapa*.

## 6. Patents

High GSL content DH lines have been patented with “variety protection right” under Korean Patent Law and Seed Industry Law. Patent IDs are as follows: BrYSP_DH005, 10-2020-0130241; BrYSP_DH014, 10-2020-0130238; BrYSP_DH016, 10-2020-0130239; BrYSP_DH017, 10-2020-0130240; and BrYSP_DH026, 10-2020-0130242.

## Figures and Tables

**Figure 1 ijms-22-07301-f001:**
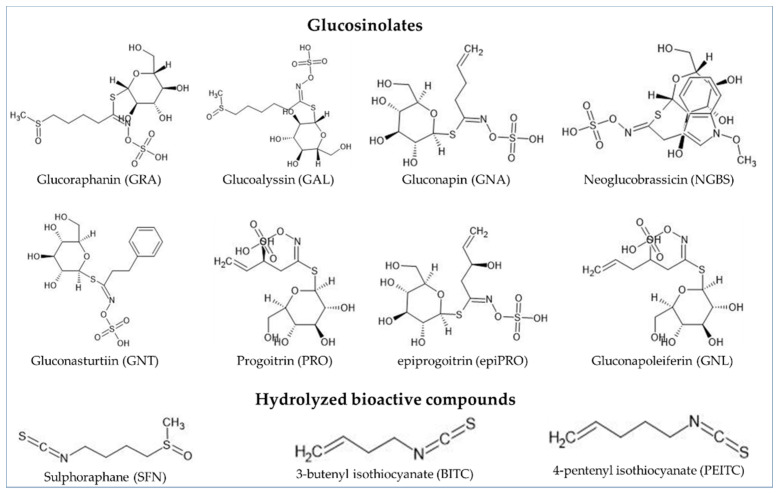
Chemical structure of glucosinolates and hydrolyzed bioactive compounds. Structure was drawn in ChemSketch software using smile notation from ChemSpider (www.chemspider.com (accessed on 7 June 2021)) database.

**Figure 2 ijms-22-07301-f002:**
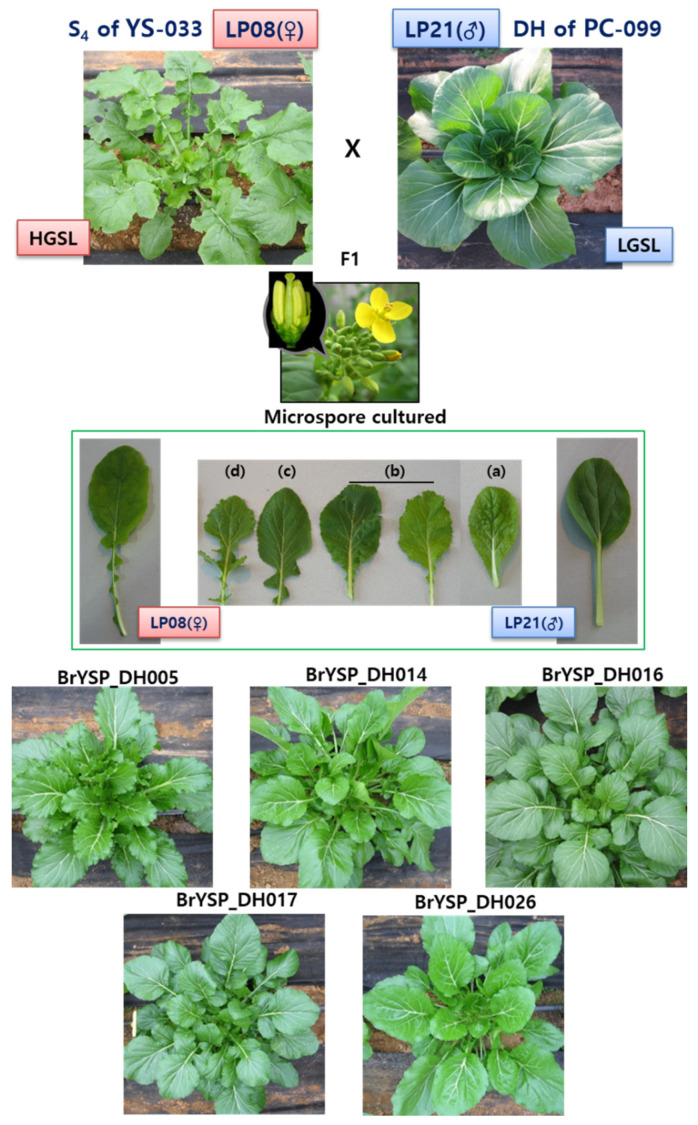
Development strategy of high glucosinolate synthesis lines using hybrid plants from different *B. rapa* subspecies.

**Figure 3 ijms-22-07301-f003:**
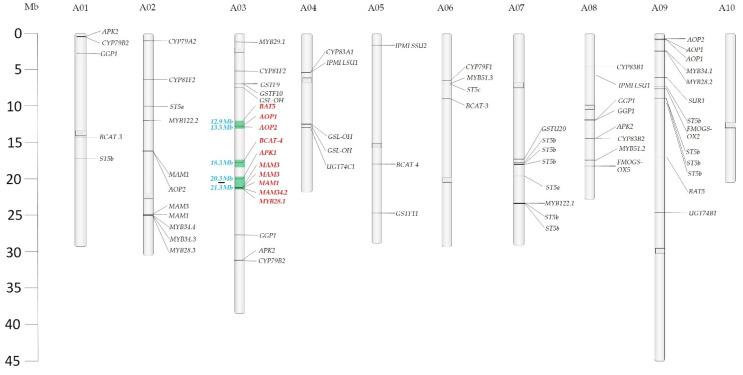
Glucosinolate (GSL) biosynthesis genes identified from recombinant blocks between LP08 and LP21. Key genes uniformly present as LP08 type in high GSL lines are indicated in red. Chromosomal regions are highlighted in the green boxes.

**Figure 4 ijms-22-07301-f004:**
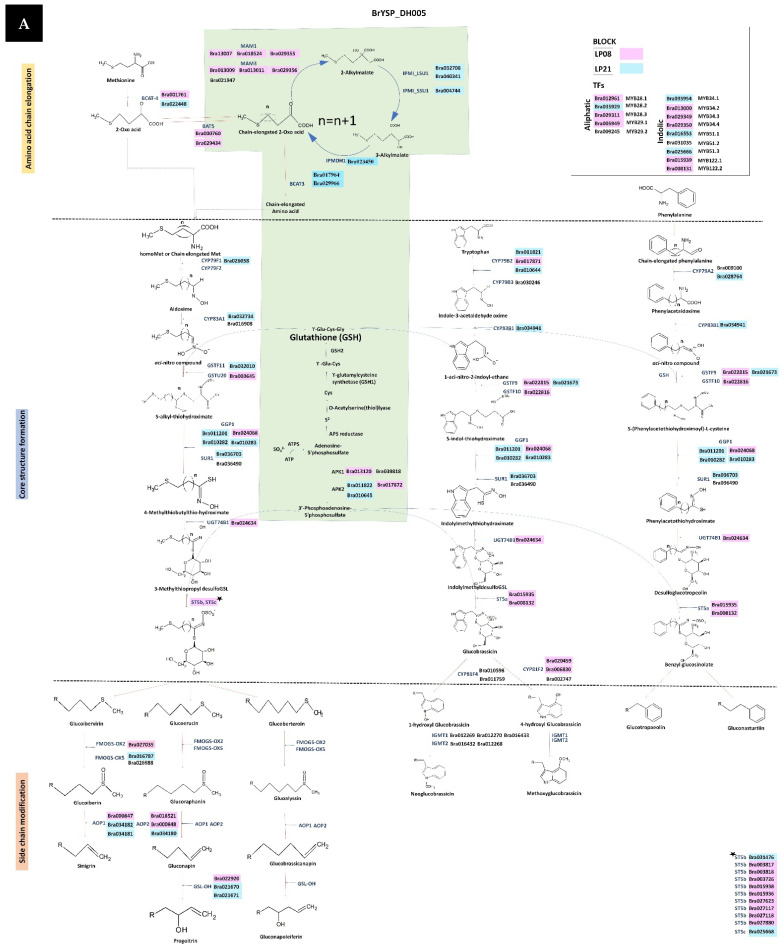
Metabolic pathway profiling of genes derived from recombinant blocks between high and low GSL parents of doubled haploid lines BrYSP_DH005 (**A**) and BrYSP_DH059 (**B**). The pathway and product structures are adapted from previously published reports [6,18,21,37] and the Brassica database BRAD. Genes not identified in recombinant blocks are not highlighted. Structure of all products were drawn using ChemSketch software.

**Figure 5 ijms-22-07301-f005:**
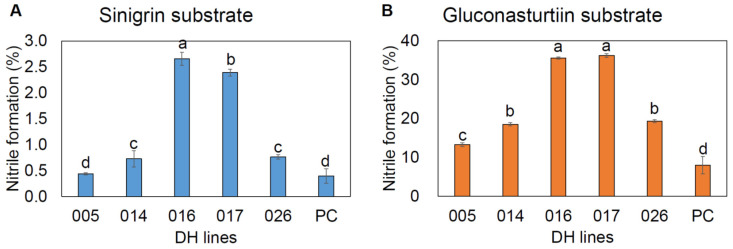
Percentage of nitrile formation in high GSL DH lines using sinigrin (**A**) and gluconasturtiin (**B**) as substrate. Lowercase letters above the error bar indicate significant differences among the accessions as determined by Tukey’s HSD test at *p* < 0.05.

**Figure 6 ijms-22-07301-f006:**
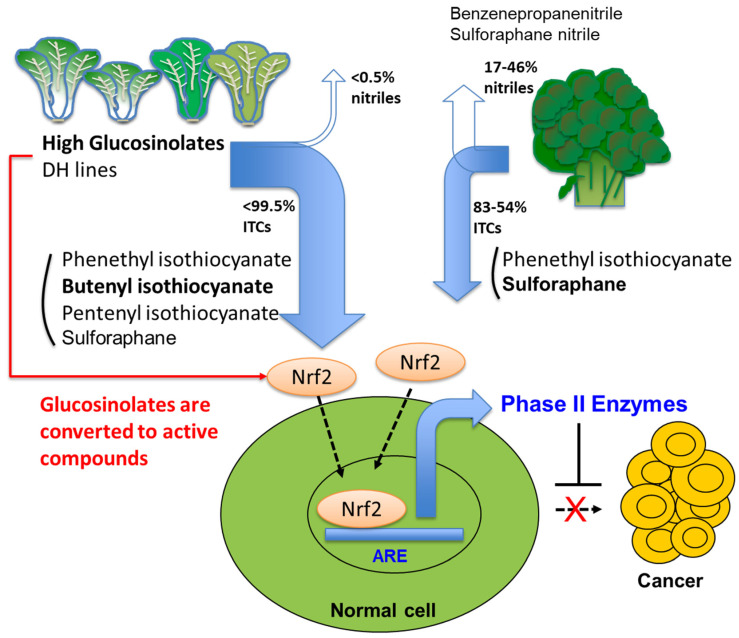
Comparison of cancer-preventive effect of high glucosinolate DH_line with low nitrile formation ability with broccoli as well-known cancer fighting vegetable as model. Genome-assisted precision breeding in *B. rapa* achieved high GSLs DH line that directly contribute to high ITCs-mediated restoration of Nrf2/ARE signaling.

**Table 1 ijms-22-07301-t001:** Total and individual GSL content (µmol·g^−1^·dw) in leaves of Chiffu, a parent of LP08 and LP21, F1, and eight BrYSP_DH lines of *Brassica rapa*.

Genotype		Aliphatic							Indolic			Aromatic	
		GRA	GAL	SNG	GNA	GBN	GNL	PRO	GBS	4-MOGBS	NGBS	GNT	Total μmol g^−1^ dw
	Chiffu	ND	ND	ND	0.00 e	0.00 f	0.00 c	0.38 ± 0.00 def	4.00 ± 0.26 a	1.09 ± 0.04 a	3.16 ± 0.52 b	0.14 ± 0.09 de	8.77 ± 0.69 ef
	LP08(♀)	0.11 ± 0.05 b	0.55 ± 0.05 g	0.17 ± 0.00 a	43.40 ± 3.63 b	0.66 ± 0.06 e	0.00c	0.71 ± 0.06 cd	0.40 ± 0.01 d	0.31 ± 0.04 c	0.76 ± 0.06 d	0.47 ± 0.14 bc	47.55 ± 3.66 bc
	LP21(♂)	0.00b	0.89 ± 0.10 def	0.00e	1.82 ± 0.66 d	0.00f	0.27 ± 0.03 a	0.88 ± 0.09 c	2.03 ± 0.02 b	0.63 ± 0.03 b	3.44 ± 0.24 b	0.81 ± 0.09 a	10.76 ± 0.36 e
	F1	0.10 ± 0.01 b	0.85 ± 0.12 ef	0.00e	1.97 ± 0.16 d	1.68 ± 0.14 d	0.34 ± 0.03 a	3.86 ± 0.29 a	0.59 ± 0.02 c	0.64 ± 0.03 b	4.58 ± 0.16 a	0.47 ± 0.05 bc	15.07 ± 0.79 d
High GSL lines	BrYSP_DH005	1.23 ± 0.18 a	3.47 ± 0.18 a	0.05 ± 0.01 cde	35.51 ± 2.84 c	0.63 ± 0.04 e	0.10 ± 0.04 b	0.22 ± 0.03 ef	0.14 ± 0.00 e	0.06 ± 0.00 f	2.03 ± 0.20 c	0.68 ± 0.34 ab	44.12 ± 2.86 c
	BrYSP_DH014	0.07 ± 0.02 b	2.16 ± 0.05 b	0.05 ± 0.01 cde	45.19 ± 2.69 ab	7.13 ± 0.62 a	0.02 ± 0.01 bc	0.06 ± 0.02 f	0.11 ± 0.01 e	0.10 ± 0.01 ef	0.77 ± 0.04 d	0.39 ± 0.04 cd	56.06 ± 3.28 a
	BrYSP_DH016	0.12 ± 0.07 b	0.43 ± 0.05 g	0.11 ± 0.07 b	44.42 ± 1.34 b	1.68 ± 0.07 d	ND	0.57 ± 0.37 cde	0.20 ± 0.01 e	0.16 ± 0.01 d	0.76 ± 0.02 d	0.24 ± 0.03 cde	48.69 ± 0.95 b
	BrYSP_DH017	0.07 ± 0.01 b	0.62 ± 0.17 fg	0.08 ± 0.02 bc	45.66 ± 1.25 ab	2.70 ± 0.09 c	0.10 ± 0.11 b	0.32 ± 0.03 ef	0.19 ± 0.01 e	0.19 ± 0.02 d	0.50 ± 0.02 def	0.20 ± 0.01 cde	50.62 ± 1.51 b
	BrYSP_DH026	0.09 ± 0.02 b	2.05 ± 0.04 b	0.07 ± 0.04 bcd	48.23 ± 1.31 a	5.39 ± 0.25 b	0.03 ± 0.02 bc	0.05 ± 0.00 f	0.08 ± 0.00 e	0.11 ± 0.00 e	0.57 ± 0.05 de	0.39 ± 0.03 cd	57.04 ± 1.54 a
Low GSL lines	BrYSP_DH009	0.04 ± 0.03 b	1.16 ± 0.10 d	0.00 ± 0.00 e	0.25 ± 0.05 d	0.26 ± 0.07 ef	0.09 ± 0.03 b	0.82 ± 0.26 c	0.12 ± 0.02 e	0.10 ± 0.01 ef	0.33 ± 0.03 ef	0.08 ± 0.03 e	3.26 ± 0.62 g
	BrYSP_DH059	0.06 ± 0.01 b	1.44 ± 0.19 c	0.01 ± 0.01 de	0.32 ± 0.06 d	0.52 ± 0.08 ef	0.30 ± 0.03 a	1.97 ± 0.23 b	0.10 ± 0.02 e	0.07 ± 0.01 ef	0.38 ± 0.03 def	0.15 ± 0.02 de	5.31 ± 0.67 fg
	BrYSP_DH061	ND	1.13 ± 0.26 de	0.01 ± 0.01 de	2.35 ± 0.81 d	1.70 ± 0.64 d	0.00 ± 0.00 c	0.05 ± 0.01 f	0.05 ± 0.01 e	0.01 ± 0.00 g	0.11 ± 0.02 f	0.23 ± 0.07 cde	5.63 ± 1.76 fg
ANOVA	Sum of square	3.852	29.573	0.091	16,645	165.68	0.525	39.6	46.36	3.601	72.88	1.669	16,984
	Mean sum of square	0.350	2.688	0.008	1513.2	15.061	0.047	3.6	4.215	0.327	6.626	0.152	1544
	F value	68.22	103.25	9.09	348.84	132.28	22.81	80.11	471.38	494.16	129.53	7.5	286.69
	*p*	****	****	****	****	****	****	****	****	****	****	****	****

GRA, Glucoraphanin; GAL, Glucoalyssin; SNG, Sinigrin; GNA, Gluconapin; GBN, Glucobrassicanapin; GNL, Gluconapoleiferin; PRO, Progoitrin; GBS, Glucobrassicin; 4-MOGBS, 4-Methoxyglucobrassicin; NGBS, Neoglucobrassicin; GNT, Gluconasturtiin; ND, Not detected. Different letters indicate significant difference between the genotypes under Duncan’s test (*p* ≤ 0.05) for three individual biological replicates. F-test are significant at **** *p* ≤ 0.0001 or Nonsignificant, respectively.

**Table 4 ijms-22-07301-t004:** Hydrolysis products of high glucosinolate doubled haploid (DH) lines.

Accessions	BITC	4-PEITC	2-PEITC	SFN	Total	Fold
BrYSP_DH005	601.0 ± 16.8	47.7 ± 1.3	40.8 ± 0.8	20.24 ± 1.33	709.8 ± 16.7	5.1
BrYSP_DH014	778.8 ± 32.9	64.5 ± 3.9	25.0 ± 1.2	2.03 ± 0.15	870.3 ± 37.6	6.3
BrYSP_DH016	425.1 ± 10.0	100.1 ± 2.5	18.5 ± 0.7	2.90 ± 0.13	546.6 ± 12.5	3.9
BrYSP_DH017	281.7 ± 15.3	120.7 ± 5.3	12.7 ± 0.8	2.35 ± 0.21	417.5 ± 21.5	3.0
BrYSP_DH026	653.0 ± 20.8	94.0 ± 2.3	34.5 ± 0.5	2.39 ± 0.15	783.9 ± 23.5	5.6
pak choi©	108.9 ± 2.7	21.6 ± 0.6	8.1 ± 0.2	0.57 ± 0.04	139.2 ± 3.6	Control

pak choi©, Commercial cultivar in South Korea. BITC, 3-Butenyl isothiocyanate; 4-PEITC, 4-pentenyl isothiocyanate (brassicanapin); 2-PEITC, 2-Phenethyl isothiocyanate; and SFN, Sulforaphane. Unit, µg∙g^−1^ dw.

## Data Availability

Not applicable.

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
