# Peer review of "Influence of Genotype on High Glucosinolate Synthesis Lines of Brassica rapa"

_ijms, 2021, doi:10.3390/ijms22147301_

Round 1

Reviewer 1 Report

Abstract.

I think the first sentence request a reference. Is it recommended by the WHO?

In general, do not use abbreviations in the abstract.

  1. 19. .. resequenced…?

L 21-23: We determined 21 the regions of genetic influence of the HGSL DH lines and found gluconapin (GNA) occurred as 22 the major GSL. I DO NOT UNDERSTAND THE CONEXION IN BETWEEN THE TWO SENTENCES.

L 24-25. We also conducted GSL hydrol- 24 ysis analysis of the HGSL DH lines using Korean pak choi cultivar as a control. I DO NOT UNDERSTAND.

  1. 30. the industrial production of potential anti-cancerous metabolites. WHICH ONES?

INTRODUCTION

It is too long and should be reduced. Avoid, as far as possible, to review the topic, but go to the focus and objectives.

It is claimed that quite a number of plants have potent anticancer compounds. While reading the introduction we can conclude that the cancer is being solved by eating cruciferous plants. Change to: there are experimental evidences supporting the idea that cruciferous compounds …..

This introduction deserves a figure with the chemical structure of the mentioned compounds.

MATERIAL AND METHODS

I am far from being a plant breeder, but to me, the plant material section needs more data.

The extraction and analysis of metabolites. Provide a reference or give more details.

I am sorry, but I am not familiarized with the techniques. My idea about material and methods is give all the necessary information to reproduce the experiments.

It is not clear to me how many plants per line have been analyzed. Is there biological replicates? The experimental design and statistical analysis are not indicated.

RESULTS

  1. 141-142. How were they selected or why did you select those lines?

Which is the agronomic value of analysing F1 lines?

DISCUSSION

It is too long and is difficult to follow.

This is a quite descriptive paper, so it is difficult to have a real biological discussion.

Author Response

Review 1
Thank you for your valuable comments. The manuscript has been revised according to your suggestions.

ABSTRACT

I think the first sentence request a reference. Is it recommended by the WHO?
à We delete this sentence ‘Glucosinolate (GSL)-rich foods are highly recommended to reduce the risk of cancer and heart disease’.

In general, do not use abbreviations in the abstract.
à We delete abbreviations of ‘SNP’, ‘GNA’, and ‘GRA’.

19. .. resequenced…? L 21-23: We determined the regions of genetic influence of the HGSL DH lines and found gluconapin (GNA) occurred as the major GSL. I DO NOT UNDERSTAND THE CONEXION IN BETWEEN THE TWO SENTENCES.
à Lines 21-22: We revised the sentence as ‘In the measured GSL, gluconapin (GNA) occurred as the major substrate in HGSL DH lines’

L 24-25. We also conducted GSL hydrolysis analysis of the HGSL DH lines using Korean pak choi cultivar as a control. I DO NOT UNDERSTAND.
à We want to test an effect for commercialized proposal and this experiment conducts by use a commercial cultivar.
à Lines 23-24: We revised the sentence as ‘Hydrolysis capacity of GSL has been analyzed in HGSL DH lines with Korean pak choi cultivar as a control’. 

30. the industrial production of potential anti-cancerous metabolites. WHICH ONES?
à Line 30: We added ‘such as gluconapin and glucoraphanin’.

INTRODUCTION

It is too long and should be reduced. Avoid, as far as possible, to review the topic, but go to the focus and objectives.
à According to the reviewer comment, in introduction review to the topic were reduced as much as possible in order to focus the objectives of manuscript. 
To make the revised manuscript looks pleasant for reading, deleted sentences in introduction were not marked in track change option. 
Details of deleted or modified sentences in introduction part in unrevised MS text are Lines 34-37, 41-47, 53-60, 85-90, 96-97, 101-102, and 107-109. 
It is claimed that quite a number of plants have potent anticancer compounds. While reading the introduction we can conclude that the cancer is being solved by eating cruciferous plants. Change to: there are experimental evidences supporting the idea that cruciferous compounds
 à Recommended sentence has been added in Line 35-36.

This introduction deserves a figure with the chemical structure of the mentioned compounds.
à Line 46: According to the reviewer suggestion, Figure has been added.

MATERIAL AND METHODS

I am far from being a plant breeder, but to me, the plant material section needs more data.

à Lines 353-358: More details about plant material have been updated.

The extraction and analysis of metabolites. Provide a reference or give more details.

à Complete procedure for GSL content (Lines 362-372), hydrolysis (414-426), and nitriles (433-450) analysis has been provided.

I am sorry, but I am not familiarized with the techniques. My idea about material and methods is give all the necessary information to reproduce the experiments.

à Materials and methods section was update with more information including DNA isolation procedure (Lines 374-386) as well as we have cited our references. 

It is not clear to me how many plants per line have been analyzed. Is there biological replicates? The experimental design and statistical analysis are not indicated.

à We have use three individual biological replicates for analysis. Statistical difference was added in Table 1 with Duncan’s test (p≤0.05).

RESULTS

141-142. How were they selected or why did you select those lines?
Which is the agronomic value of analysing F1 lines?

à As doubled haploids (DH) lines were produced from the F1 which was generated between two different parents on glucosinolate content, we have tested the GSL content of F1 plants also.
à Lines used in this manuscript has been selected based on glucosinolate content and recombinant blocks identified by high-density SNP mapping.
DISCUSSION

It is too long and is difficult to follow.

This is a quite descriptive paper, so it is difficult to have a real biological discussion.

à Descriptive sentence in the discussion part was either removed or modified according to the reviewer’s suggestion. Discussion section has been reduced as much as possible to make easy in following it.
To make the revised manuscript looks pleasant for reading, deleted sentences in discussion were not marked in track change option. 
Details of deleted or modified sentences in discussion part in unrevised MS text are Lines 280-291, 296, 301-302, 309-313, 324-326, 329-331, 336, 345-347, 349-356, 359-363, 373-378, 383-386, 390-392, 397-400, 432-434, 443-453, and 459-465.

Reviewer 2 Report

The ms “Influence of Genotype on High Glucosinolate Synthesis Lines of Brassica rapa” describes an interesting case of bulked segregant analysis to study the genetic effect of chromosome regions containing genes for glucosinolate biosynthesis on accumulation of these compounds in Brassica rapa plants.

There are some aspects to be improved.

The first one is about Materials and Methods which need to be better detailed. Some details have to be added about the final set of SNPs used for the analysis. What filtering has been carried out to obtain high-quality SNPs? What about filtering based on missing data, minor allele frequency and so on?

In Results section, a Table with results of ANOVA should be added.

An important aspect is that the BSA analysis has been carried out with recombinant blocks containing genes related to GLS biosynthesis, but do these regions explain all the observed phenotypic variation for GLS accumulation? A more general mapping should be carried out with the genome-wide SNPs to identify all of the chromosomal regions explaining the traits, not only those regions containing GLS related genes.

Author Response

Review 2
Thank you for your valuable comments. The manuscript has been revised according to your suggestions.

The ms Influence of Genotype on High Glucosinolate Synthesis Lines of Brassica rapa??describes an interesting case of bulked segregant analysis to study the genetic effect of chromosome regions containing genes for glucosinolate biosynthesis on accumulation of these compounds in Brassica rapa plants.

There are some aspects to be improved.

The first one is about Materials and Methods which need to be better detailed. Some details have to be added about the final set of SNPs used for the analysis. What filtering has been carried out to obtain high-quality SNPs? What about filtering based on missing data, minor allele frequency and so on?

à Reads were converted from the binary base call (BCL) format file using bcl2fastq V2.20 software with parameter ‘–barcode-mismatches 0’. To remove low-quality bases, clean reads were trimmed using sickle v1.33 with a phred quality score threshold of 20 (accuracy 99%) to derive high-quality reads.

In Results section, a Table with results of ANOVA should be added.

à We have updated the Table 1 with statistical difference using Duncan’s test (p≤0.05).

An important aspect is that the BSA analysis has been carried out with recombinant blocks containing genes related to GLS biosynthesis, but do these regions explain all the observed phenotypic variation for GLS accumulation? A more general mapping should be carried out with the genome-wide SNPs to identify all of the chromosomal regions explaining the traits, not only those regions containing GLS related genes.

Figure with complete chromosome mapping have been added as supplementary figure 1.

à Supplementary Figure 1. General mapping of entire chromosome of doubled haploid lines for bulked segregant analysis of A) high GSL lines, and B) low GSL lines. a, Circular representation of pseudomolecules. b, Histogram representation of GC rate for each 100kb window. c, Histogram representation of gene density for each 100kb window. Recombinant block of BrYSP_DH005, DH014, DH016, DH017, DH026, and DH009, DH059, and DH061 lines in respective to LP08 and LP021 parents.

Round 2

Reviewer 1 Report

Thanks for rising my comments to the previous version

Author Response

Thanks to your revision.

Reviewer 2 Report

Actually, I do not see in the ms most of the requested revisions:

  • Table 1 was updated with statistical difference, but the anova table is still missing
  • Results on general mapping have to described in the text: are there chromosomal regions associated with the phenotype which do not contain biosynthetic genes? These could contain regulator genes.

Author Response

Review 2
Thank you for your valuable comments. The manuscript has been revised according to your suggestions.

Table 1 was updated with statistical difference, but the anova table is still missing.
à Table 1 was updated with details of ANOVA.

Results on general mapping have to described in the text: are there chromosomal regions associated with the phenotype which do not contain biosynthetic genes? These could contain regulator genes.
à We developed 161 BrYSP_DH lines and were going to survey a lot of phenotype traits of seed coat color, flowering times, GSL contents, locules formation and leaf edge degree etc. This manuscript is focus on HGSL plant genotypes on parents.